# Application of Hydrogel Spacer SpaceOAR Vue for Prostate Radiotherapy

**Satvik R. Hadigal and Atul K. Gupta ***

Department of Radiology, Rochester General Hospital, 1425 Portland Ave, Rochester, NY 14621, USA
* Correspondence: atul.gupta@rochesterregional.org

**Abstract:** Damage in the surrounding structures, including the rectum, due to unintended exposure to radiation is a large burden to bear for patients who undergo radiation therapy for prostate cancer. The use of injectable rectal spacers to distance the anterior rectum from the prostate is a potential strategy to reduce the dose of unintended radiation to the rectum. Hydrogel spacers are gaining increasing popularity in the treatment regimen for prostate cancer. After FDA approval of SpaceOAR, specialists are receiving an increasing number of referrals for hydrogel placements. In this paper, we review hydrogel spacers, the supporting clinical data, the best practices for hydrogel placement, and the risk of adverse events.

**Keywords:** SpaceOAR; SpaceOAR Vue; hydrogel spacer; rectal toxicity; prostate cancer; prostate radiation

## 1. Introduction

Prostate cancer is the most common cancer diagnosed in male patients in the United States [1]. The majority of these patients present with localized or regional disease, and a vast majority of this group may be eligible for curative treatment with radiotherapy. Since the 10-year survival rate for prostate cancer exceeds 80%, most men will survive their disease and be at risk for experiencing negative consequences from radiotherapy [2]. Well-known side effects of radiotherapy are acute and chronic toxicity. Acute toxicity tends to be mild and self-limiting; however, chronic toxicity (e.g., urinary dysfunction, sexual dysfunction, bowel dysfunction, rectal bleeding, fistula formation, and tissue necrosis) can be debilitating and morbid [3]. Although sophisticated radiation techniques such as intensity-modulated radiotherapy (IMRT) and proton beam therapy (PBT) have been implemented to alleviate rectal toxicities, they do not completely eliminate the toxicity. Rectal spacers are an attractive solution that separate the posterior aspect of the prostate from the anterior rectal wall. In this article, we review SpaceOAR[TM] hydrogel and iodine-containing SpaceOAR Vue[TM] hydrogel, including clinical data supporting these hydrogels, adverse events from hydrogel insertion, and appropriate hydrogel placement.

## 2. Background

The prostate is located in the pelvis and bordered posterosuperiorly by the seminal vesicles and posteriorly by the rectum. The posterior boundary of the prostate is surrounded immediately by the prostatic fascia, followed by Denonvilliers' fascia, which is also called the posterior pelvic fascia or seminal vesicle fascia. Denonvilliers' fascia is a single fused fascial layer composed of dense collagen, smooth muscle, and coarse elastic fibers [4]. Moving more posteriorly, there is a loose, areolar adipose tissue of the mesorectum, followed by the muscular layers of the rectal wall. The loose areolar tissue acts as a potential space and is easy to dissect and separate. All the layers mentioned above separate the prostate and the rectum by 2–3 mm [5]. This meager distance makes radiation treatment significantly challenging, as most cancers develop in the peripheral

zone located posteriorly in the prostate gland [6]. The rectum is a dose-limiting critical structure in radiation treatment and is referred to as the primary Organ at Risk (OAR) in prostate radiotherapy [7].

### 3. Prostate Radiation Treatment

There are two main types of radiation therapy for prostate cancer, external beam radiation therapy (EBRT) and internal radiation therapy or brachytherapy (BT). There are various types of external beam radiation therapies, a few of which are intensity-modulated radiotherapy (IMRT), image-guided radiotherapy (IGRT), stereotactic radiotherapy (SBRT), and volumetric arc radiotherapy (VMAT). IMRT uses nonuniform radiation beam intensities to target tumors. IGRT is a process that integrates tumor positioning, image guidance tools, and other motion management systems to better direct the radiation beam to the tumor [8]. SBRT is a fusion of state-of-the-art tumor imaging with precision radiation treatment delivery systems and delivers a complete course of radiation in a shorter amount of time and in fewer visits when compared to IMRT [8]. On the other hand, brachytherapy is divided into two main forms: low-dose brachytherapy (LDR) and high-dose brachytherapy (HDR) [9]. Primary or scatter radiation from any of these therapies inadvertently puts neighboring organs at risk for toxicity, with the rectum being one of the vital organs at risk [3]. In radiation oncology, specification of the volumes is crucial in the planning and evaluation of the patient's treatment in EBRT. Gross tumor volume (GTV) refers to the volume of known tumor that is imaged. Clinical target volume (CTV) is defined as the volume that represents the known imaged tumor and/or the subclinical malignant disease that is not imaged. During treatment, the volume delineated around the tumor that ensures a prescribed dose will actually be delivered to all parts despite geometrical uncertainties such as organ motion and setup variations is called the planning target volume (PTV) [8].

### 4. Spacers

Unintentional radiation to adjacent organs can be reduced by either decreasing the dose of radiotherapy or by creating separation between the target organ and its surrounding structures. Decreasing the dose is achieved with fractionation, which reduces but does not eliminate the risk of rectal toxicity [10]. Increasing the distance between the prostate and the rectum decreases radiation toxicity to the surrounding OAR. For external beam radiotherapy, the distancing provides better PTV during EBRT. An average of 1.26 cm of perirectal spacing relatively decreases the rectal volume by 73.3% after receiving at least 70 Gy (rV70) [11]. On the other hand, for BT where the source of radiation is inside the prostate, increasing the distance between the prostate and the rectum decreases the radiation dose by the square of the distance. The perirectal space is usually around 2–3 mm in thickness, and hydrogels can potentially create up to 1.5 cm of separation [5]. Among the various space-creating solutions that have been developed (e.g., bioabsorbable balloon, human collagen, hyaluronic acid, and polyethylene glycol (PEG)-based hydrogels), hydrogels have the largest wealth of supporting clinical data and are the most widely used [12].

SpaceOAR$^{TM}$ hydrogel (Boston Scientific, Marlborough, MA, USA) is an absorbable polyethylene glycol hydrogel spacer that received FDA approval in 2015 [13]. Composed of biodegradable polyethylene glycol, this hydrogel maintains the space for the 3 months during treatment, after which it spontaneously breaks down by hydrolysis and is excreted renally after 6 months [5]. This hydrogel has been used in clinical trials using IMRT and SBRT to treat prostate cancer and successfully demonstrated protection to the rectum [5,14]. The preferred approach to IMRT and SBRT treatment planning traditionally uses MRI to visualize the rectum, as the tissue interface between the posterior prostate and anterior rectum is better determined by MRI than by CT imaging. MRI has shown a superior definition of the prostatic borders and reduces the clinical target volume (CTV) by 30% when compared to CT imaging alone [15,16]. With the visual aid of a hydrogel, which is relatively T2 hyperintense, the anatomical separation obtained significantly decreases

the rectal toxicity with a mean rectal V70 of 3.3% compared to 12.4% without hydrogel ($p < 0.0001$) [5].

There is physiological movement of the prostate and rectum between treatment days (interfractional motion) and within the same treatment day (intrafractional motion) that affects the CTV and PTV margins, which can lead to an increased dose delivered to healthy tissue and possibly the surrounding OAR [17,18]. Hydrogel spacers have demonstrated a significant impact in lessening prostate movement in the anteroposterior rotational and translational shifts ($p = 0.033$). Furthermore, there was a positive impact on dampening the anteroposterior translational shift, albeit not significant ($p = 0.07$) [19]. Nevertheless, despite MRI image localization and the dampening effects of hydrogel, fiducial markers are recommended and are currently the standard treatment. They can help localize the prostate during treatment sessions and are used to match the original position determined by planning the CT. One major downside to the SpaceOAR$^{TM}$ hydrogel is that the radiodensity of this hydrogel is similar to soft tissues, such as the prostate and the rectum. Consequently, these rectal spacers are difficult to visualize on CT scans, which can make the contouring accuracy of the prostate and rectum challenging. This can potentially lead to higher inadvertent radiation to the surrounding structures during the planning phase if CT is the only modality available [20]. Due to the intrafractional and interfractional motions of the prostate, the treatment plan can be altered between the planning phase and in-treatment images based on changes in patient anatomy with a kV cone beam CT (CBCT) [20]. A newer product, SpaceOAR VUE$^{TM}$ with iodinated contrast and enhanced visibility on CT, can improve the prostate/rectum contouring accuracy during the planning and visualization of the target region during treatment so that consistent therapy can be administered. Previously, when prostate cancer patients had contraindications to MRI imaging, such as pacemakers, implantable cardioverter defibrillators, metallic foreign bodies, cochlear implants, or intracranial aneurysm clips, SpaceOAR$^{TM}$ was not an option.

SpaceOAR Vue$^{TM}$ hydrogel was developed to include an iodinated crosslinked PEG. This innovation causes the spacer to appear radiopaque, so that the hydrogel can be visualized on CT [21]. The hydrogel is covalently bonded with iodine to avoid free-floating molecules in the body during degradation to prevent allergic reactions to iodine. The properties of the standard SpaceOAR$^{TM}$ (e.g., transformation to solid from liquid components in fractions of a second, stability over 3 months during radiotherapy, and eventual clearance by the body) are unchanged in the iodinated version [22]. Same as its precursor, SpaceOAR Vue$^{TM}$ provided comparable dosimetric consistencies and relative prostate-to-rectal separation, despite a smaller average delineated volume. Noniodinated hydrogel measured significantly higher in volume at 10.6 versus 8.9 mL ($p < 0.001$) when compared to iodinated hydrogel on CT likely due to over-contouring from the inability to accurately demarcate the anatomic boundaries. This was hypothesized to be due to incorrect MR/CT fusion techniques, which is exacerbated when the preprocedural MRI and CT are taken on separate days [20].

## 5. Clinical Data for Hydrogels

### 5.1. IMRT

A single, prospective, multicenter phase III randomized trial of dose-escalated image-guided IMRT using a hydrogel spacer showed a significant decrease in radiation to the rectum. At a 15-month follow up, rectal toxicity and urinary toxicity were significantly reduced when measured by physicians. In addition, the Expanded Prostate Cancer Index Composite (a validated instrument that measures patient-reported health-related quality of life after prostate cancer treatment) demonstrated improved bowel quality of life (QOL) in patients for whom a hydrogel spacer was used [5]. Reassuringly, many differences measured at 15 months were maintained or increased at the 3-year follow-up [23]. The use of a hydrogel spacer also decreased the incidence of bother secondary to urinary frequency, with the control arm measuring 18% and hydrogel arm measuring 5% ($p < 0.05$). In addition, there was a statistically significant improvement in the average urinary QOL at the 3-year

follow-up favoring the spacer arm by +0.6 points versus −3.3 points when compared with the control arm ($p < 0.04$) [23]. Furthermore, the use of a hydrogel spacer decreased the radiation toxicity in the penile bulb; this was associated with improved erectile function compared with the control group based on patient-reported sexual QOL [24].

A secondary analysis was performed on patients from the phase III trial mentioned above. The objective was to identify a subgroup of patients who may not benefit from spacer placement based on clinical, anatomic, and dosimetric factors. Based on this study, it had previously been suggested that only patients with a large prostate volume would benefit from spacer placement. However, there was an absolute reduction in rectal radiation after hydrogel spacer placement regardless of the prostate size. The absolute reduction in radiation for prostates sizes <40 mL and >80 mL decreased from 13% to 3% and from 12% to 2%, respectively ($p < 0.01$). Similarly, the study found that, regardless of the prostate-to-rectum distance, there was a significant decrease in the absolute rectal toxicity with the placement of a hydrogel spacer. When the mid-prostate gland-to-rectal space measured 0, then there was an absolute reduction in the rectal toxicity from 12.4% to 3.2% ($p < 0.01$) after hydrogel spacer placement. This absolute reduction remained significant, decreasing from 12.2% to 2.0% when the mid-prostate gland-to-rectal space was >2.2 mm prior to the hydrogel spacer ($p < 0.01$). The study also assessed the associations between prior abdominal, pelvic, and hemorrhoid surgery on rectal toxicity and found no significant correlation with the baseline bowel QOL ($p = 0.8$) [25]. This suggests that all patients undergoing IMRT, irrespective of prostate size, intrinsic anatomic distance, and prior surgeries, could benefit from hydrogel spacers.

*5.2. SBRT*

Fewer and larger doses (hypofractionated) of SBRT radiotherapy improve the cost and patient convenience relative to conventional fractionated radiotherapy [26]. Nevertheless, studies have demonstrated substantial genitourinary and gastrointestinal toxicity in patients undergoing aggressive regimens of dose-escalated hypofractionated SBRT [27]. Although these regimens achieved high rates of freedom from biochemical failure, increased rectal toxicity was observed. For example, one study with 91 patients who received dose-escalated 45–50 Gy in 5 fractions demonstrated that all developed rectal ulcers in the anterior rectal wall, although they eventually resolved [14,28]. Furthermore, 5/91 patients developed a rectourethral fistula requiring a colostomy [28]. A systematic review showed that, when a hydrogel was used, patients who underwent dose-escalated SBRT regimens (37.5–45 Gy in 5 fractions) demonstrated a low risk of late grade ≥ 2 GI toxicity [29]. Regardless of the total radiation dose utilized, rectal radiation exposure was decreased by 29–56% across the measured dosimetric profile curve, represented as a percentage of the maximum prescribed radiation dose when rectal spacers were used [29]. A Multi-Institutional Phase 2 Trial of High-Dose SAbR (stereotactic ablative radiotherapy) (45 Gy in 5 fractions) using hydrogel demonstrated a significant reduction in the incidence of rectal ulcer. A rectal ulcer rate of 14.3% (95% CI, 6.0–27%; $p < 0.001$) was observed by direct anoscopy in low-risk and intermediate-risk prostate cancer when compared to 100% from the prior phase 1/2 trial results (90% power; $\alpha = 0.05$ in a 2-sided exact binomial test). Moreover, no subsequent grade ≥ 3 GI toxicity was observed with the hydrogel spacer when compared to 7% of the patients without a spacer [14,28].

In volumetric-modulated arc therapy (VMAT) prostate stereotactic body radiotherapy (SBRT), dose coverage of the planning target volume is challenging when attempting to spare the rectum, bladder, and urethra. Utilizing hydrogels has demonstrated improvement in target dose coverage and rectal radiation sparing [30].

Previously, urethrogram-directed SBRT was used in patients with contraindications to MRI [31]. Although a CT urethrogram aids in the identification of the prostatic apex, there is increased uncertainty in the location of the anterior rectal wall with respect to the prostate when using urethrogram-based treatment planning without MRI fusion assistance. This subset of patients has the potential to benefit from the use of SpaceOAR Vue™ iodinated

spacers to mitigate the risk of GI toxicity [32]. The iodinated hydrogel is easily visualized on CT and helps delineate the rest of the prostate–rectum interface for better-targeted SBRT.

*5.3. Brachytherapy*

Brachytherapy is an accepted single-modality treatment for low-risk and favorable intermediate-risk prostate cancer or as part of a combination regimen for unfavorable intermediate- and high-risk prostate cancer [33]. Low-dose rate (LDR) brachytherapy yields higher biochemical progression-free survival rates when compared to treatment regimens with EBRT and prostatectomy [34–36]. Rectal toxicity after brachytherapy has been reported to be as high as 39% and is most likely due to the proximity of the rectum from the seeds implanted within the prostate gland [37]. Taggar et al. demonstrated successful placement of the hydrogel after seed implantation during the same procedure and showed that the hydrogel placement reduced the measured radiation dose to the rectum and demonstrated decreased acute rectal toxicity [38].

SpaceOAR Vue<sup>TM</sup> offers the added clinical benefit of post-implant contouring and analysis, which is particularly advantageous in the context of LDR brachytherapy [39]. Detailed contouring of the anterior rectal wall and posterior aspect of the prostate is essential for accurate dosimetry, which is most often calculated based on CT imaging alone. A noniodinated hydrogel, in the presence of edema and bleeding around the prostate, can demonstrate proper contouring and accurate post-implant dosimetry challenging. Furthermore, the streak artifact caused by the brachytherapy seeds can obscure the boundaries of the rectal wall. The iodinated crosslinked PEG component of SpaceOAR Vue<sup>TM</sup> hydrogels improves visualization in CT imaging and is therefore beneficial in LDR brachytherapy.

## 6. Placement

An endorectal ultrasound probe containing a side firing and an end firing is required to allow visualization of both the axial and sagittal planes for accurate placement. An appropriate table for placing the patient in the dorsal lithotomy and a stand for an ultrasound probe are highly encouraged for patient and physician comfort during the procedure. For preparation, patients are instructed to do a Fleet enema the night before for accurate visualization when using the endorectal ultrasound probe and remain NPO (nothing by mouth) for 8 h prior to the procedure for moderate sedation. The risk of infection is much lower than transrectal procedures, and hence, prophylactic antibiotics are not typically used [40]. Patients are preferably given moderate sedation in the preprocedural holding to alleviate anxiety.

Once the patient is positioned in the dorsal lithotomy position, a rectal examination is performed with lidocaine jelly to gently dilate and relax the anal sphincter. The perineum is prepared with a chlorhexidine scrub and draped using a sterile technique. An end and side firing endorectal ultrasound probe, prepared with a probe cover and gel, is introduced into the rectum to visualize the prostate in the axial and sagittal planes. Other anatomical landmarks, such as the rectal hump, seminal vesicles, and perirectal fat, should be identified prior to proceeding further.

The following describes the hydrogel preparation for SpaceOAR Vue<sup>TM</sup> and contains a few minor differences in technique when compared to its precursor. The SpaceOAR Vue<sup>TM</sup> hydrogel is supplied as a dry PEG powder (yellow colored powder) that is reconstituted with a diluent (green-colored solution) with the help of a provided plastic injector (Figure 1). This mixture is shaken vigorously for 20 s and is left on the table for 5 min to fully dissolve. During this time, a 23-gauge needle with a lidocaine syringe is advanced through the perineum, approximately 1.5 cm above the rectum. Under US guidance, using the sagittal viewing plane, the needle is advanced as lidocaine is injected into the anticipated trajectory towards the mesorectum. Once the powder is dissolved, 5 mL of the mixed solution is withdrawn into a 10-mL syringe, and any excess is discarded. Then, 2 mL of air is withdrawn into the syringe for airlock. Similarly, 5 mL of accelerant is withdrawn into a 10-mL syringe, the excess is discarded, and 2 mL of air is withdrawn into the syringe

for an airlock. Both syringes are connected to the supplied "Y" connector and syringe holder. A provided plastic connector is inserted over the thumb rests of both the plungers and maintains equal volumes of the syringes. This entire apparatus should be held in the upright position with air in the hub and needle end of the barrel. This prevents inadvertent mixing of the two solutions and causing occlusion of the delivery apparatus.

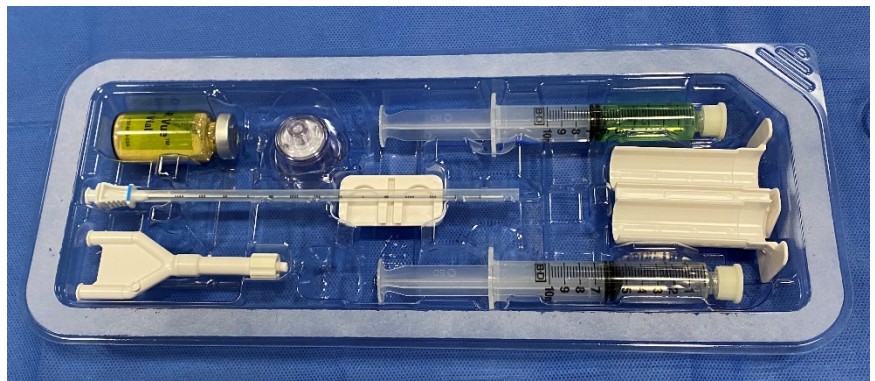

**Figure 1.** Contents of the SpaceOAR Vue. Dry PEG powder (yellow-colored powder, **top left**) is reconstituted with a diluent (green-colored solution, **top right**). A Y-connector (**bottom left**) and plastic syringe holder (**middle right**) connects the accelerant syringe (**bottom right**) to the PEG powder syringe.

An 18-gauge needle is flushed with saline to expel any air and advanced under ultrasound guidance to the mesorectal fat via a transperineal approach, taking care to use the similar lidocaine tract to minimize patient discomfort. Care should be taken to ensure the bevel is pointing down (toward the rectum). Once the prostate mid-gland is reached on the sagittal plane, the axial viewing plane should be utilized to confirm the needle position. This visualization and movement of the needle laterally is performed to ensure the injection is done midline. Aspiration via the needle should not produce any blood. Injection of 1 to 2 mL of saline is used for hydrodissection and should fill the target zone, followed by quick dissipation. Placement of the needle in the correct plane and adequate hydrodissection are critical components of the procedure to ensure high-quality spacer positioning without the complication of injection into the rectal wall or prostate [41].

Gently unscrew the saline syringe while keeping the needle in the same position and attach the two syringes provided by the kit. The airlock is expelled to the level of the shoulders of the syringe. Do not introduce air, which may distort the ultrasound image. The hydrogel is injected through the 18-gauge needle over 10–12 s under ultrasound guidance and withdrawn after completion. Stopping during injection may result in device plugging, requiring the preparation of a replacement system [42]. The mixture of SpaceOAR Vue^TM is more viscous than a traditional hydrogel spacer and is encountered mostly during injection. A sagittal US view confirmation is obtained (Figure 2), followed by a post-procedure CT to ensure proper placement (Figure 3a,b). For comparison, an axial and sagittal CT are obtained with the insertion of SpaceOAR to demonstrate the superior visualization of SpaceOAR Vue with iodinated contrast (Figure 4a,b).

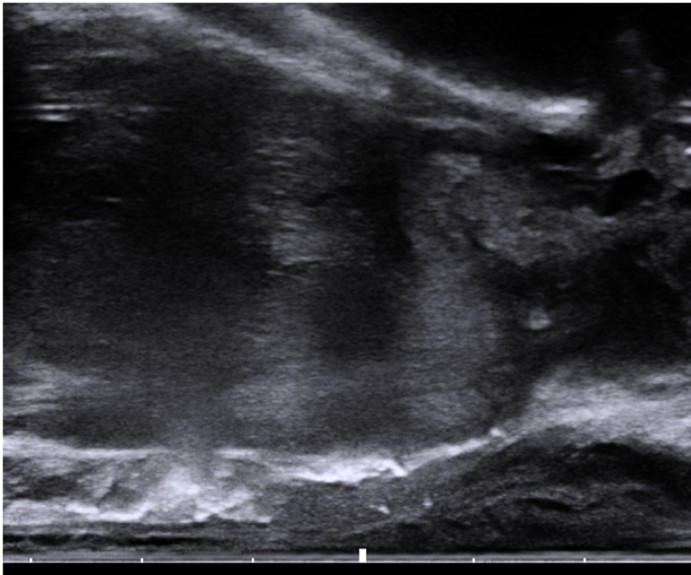

**Figure 2.** Sagittal view of the transrectal probe shows an echogenic spacer between the rectum and prostate.

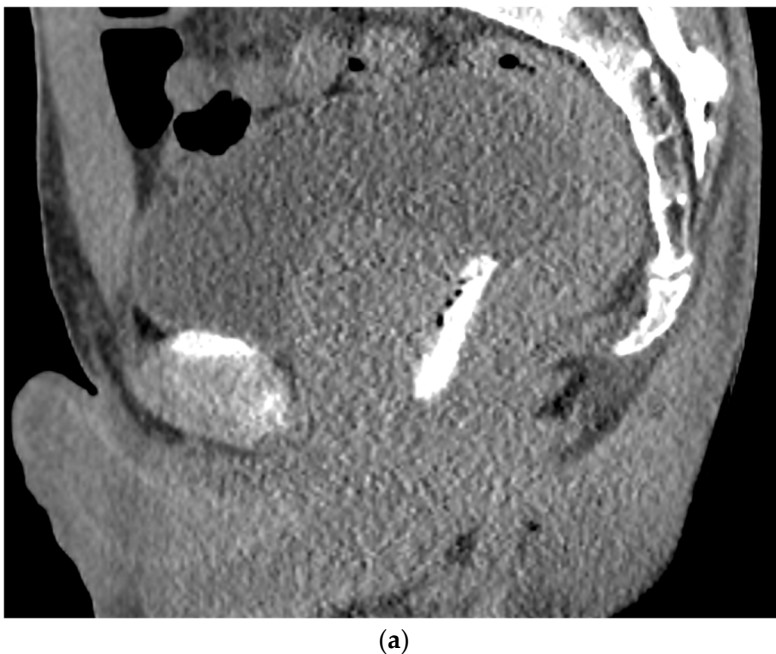

(**a**)

**Figure 3.** *Cont.*

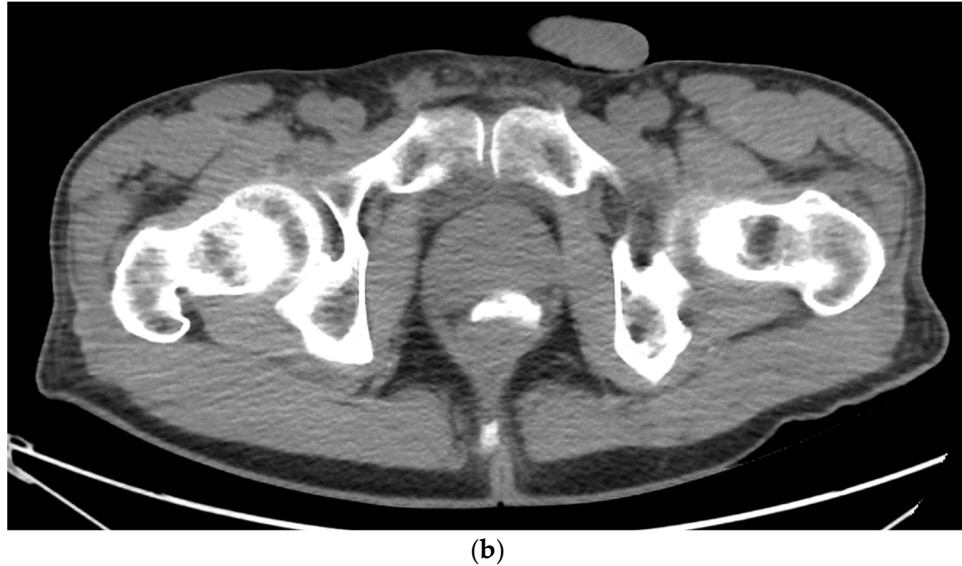

(**b**)

**Figure 3.** (**a**). Sagittal view after injection of SpaceOAR Vue shows 1 cm of distance gained between the rectum and prostate at the level of the mid-gland. (**b**). Axial view after injection of SpaceOAR Vue shows 1.1 cm of distance gained between the rectum and prostate at the level of the mid-gland.

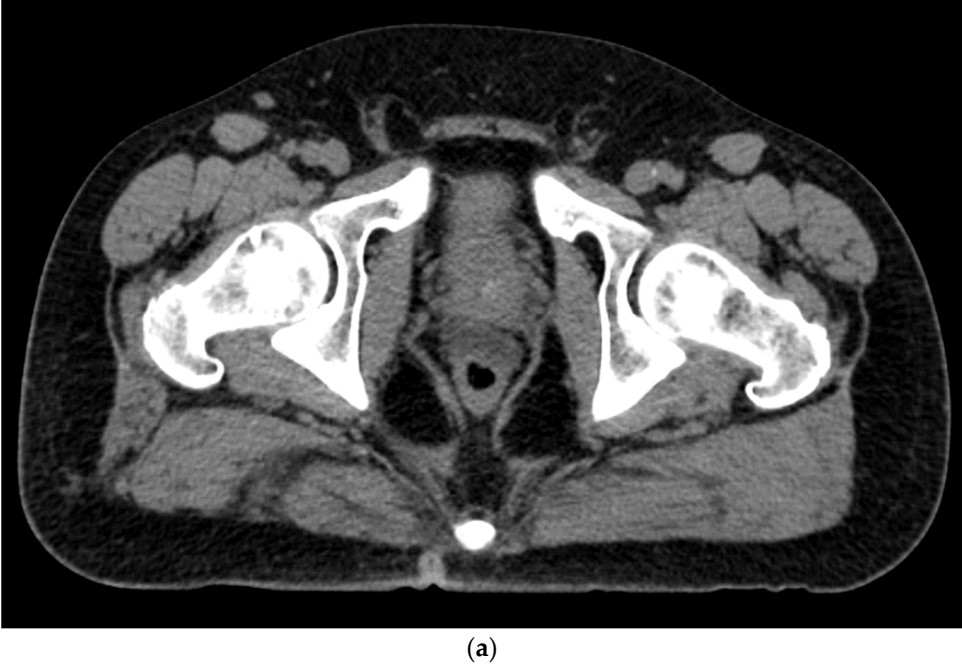

(**a**)

**Figure 4.** *Cont.*

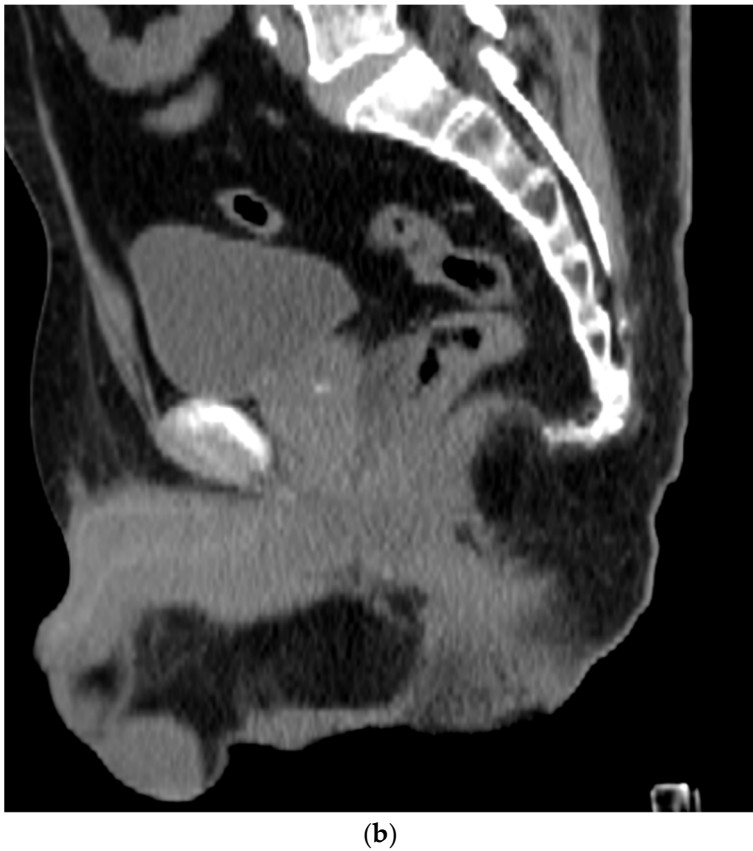

(**b**)

**Figure 4.** (**a**). Axial view after the injection of SpaceOAR shows 1.1 cm of distance added between the rectum and the prostate at the level of the mid-gland. Due to the lack of iodine, the hydrogel is difficult to visualize and is seen as a hypodense soft tissue density. (**b**) Sagittal CT shows a soft tissue hypodense SpaceOAR between the rectum and prostate.

## 7. Contraindications to Hydrogel Placement

As per the manufacturer, there are no explicit contraindications to the placement of SpaceOAR™ or SpaceOAR Vue™ [21]. Therefore, all listed contraindications here are a relative contraindication and should be carefully explored on a case-by-case basis. For example, the potential of iodine to cause an allergic reaction is a possible adverse effect associated with SpaceOAR Vue™ hydrogels. Theoretically, this is avoided by the covalent bond between the iodine and the PEG, as no free iodine molecules are available to mount an allergic reaction. However, iodine sensitivities or allergies have not been extensively studied, and the company defers to the physician to assess the risks and benefits of SpaceOAR Vue™ use in patients with a documented allergy [21].

Patients with a history of pelvic surgery or pelvic radiation, which can cause significant scar tissue is another relative contraindication to hydrogel placement. Mahal et al. demonstrated that adequate separation between the prostate and the rectum could be achieved with hydrogel placement in previously irradiated prostate cancer patients [43]. Compared to the average of 12.0 mm in patients with no radiation, there was a separation measuring 10.9 mm in patients with prior EBRT and a separation of 7.7 mm for patients with prior brachytherapy [43]. Thus, spacer injection can be considered in patients with a history of pelvic radiation at the discretion of the physician; however, further investigation/evidence would be beneficial in making this determination.

Special consideration should be given when considering hydrogel spacer use in patients with posterior extracapsular extension through Denonvilliers' Fascia. Some in the radiation oncology community believe that cancer cells are pushed away from the radiation field and should therefore be avoided. However, the tumor abutting the capsule is acceptable for hydrogel placement [44].

Finally, ongoing pelvic infections, including prostatitis, is a relative contraindication, as hydrogels may act as a surface and reservoir for microbes.

## 8. Adverse Effects of Hydrogels

The SpaceOAR Vue[TM] needle should be inserted under ultrasound guidance to maintain needle tip visibility and prevent rectal wall penetration. In a study including 258 patients, only 1.6% (i.e., four patients) experienced rectal wall penetration [45]. If the needle enters the rectal lumen at any time, the procedure should be abandoned to avoid infection and rectal wall infiltration. A few studies have shown that the majority of cases in which hydrogel was injected in the rectal wall are resolved with conservative management and time [46]. The hydrogel is thought to slowly resorb over time. However, in a handful of cases, surgical colostomy and surgical intervention were required [46].

According to the Manufacturer and User Facility Device Experience (MAUDE) database, a few major complications included severe anaphylaxis, rectourethral fistula, abscess formation, and sepsis. Interventions such as abscess drainage, diverting colostomy, and ICU admission were required as further management. Unfortunately, two deaths were reported after hydrogel placement. One patient developed perineal abscess and subsequently passed away from alcoholic cardiomyopathy, and another patient developed dizziness/nausea post-procedure, leading to unresponsiveness and death. The former was determined as an unlikely causal relationship due to death by the patient's baseline morbidity, and the latter patient death's causal relationship was not assessable [47]. Further studies are needed to understand and gain knowledge about the potential rare and serious complications of this procedure.

## 9. Cost Effectiveness

Eventually, the long-term usability of SpaceOAR and SpaceOAR VUE will be decided by the cost effectiveness, as there is an added cost to radiotherapy, in addition to hydrogel insertion having its own complications. The Centers for Medicare and Medicaid approved the current procedural terminology code (55874) in 2018, and the reimbursement rates vary depending on the kind of facility performing the procedure. Levy et al. reported that hydrogel spacers are highly cost effective in the setting of ambulatory service care (ASC) and cost effective in a physician's office or hospital outpatient department. Using a commonly accepted willingness-to-pay (WTP) threshold in the United States of USD $100,000/quality-adjusted life year (QALY), a procedure that is highly cost-effective is considered USD <$50,000 and cost effective is considered USD <$100,000.

This study was based on patients receiving conventionally fractioned radiotherapy [48]. Other studies have arrived at a similar conclusion that hydrogel placement is cost effective, as it provides reductions in the frequency of gastrointestinal, genitourinary, and sexual dysfunction complications along with QOL improvements [49,50]. On the other hand, Hutchinson et al. demonstrated the cost effectiveness of the hydrogel spacer only in high SBRT (50 Gy) but not in low SBRT (36 Gy) and conformal RT dose escalation [51].

All the studies mentioned were performed with SpaceOAR, and no studies were performed with SpaceOAR Vue. SpaceOAR Vue is more expensive than its precursor, likely due to the added cost of covalently bonding iodine to the PEG. This added expense can be justified by the financial gain of not requiring an MRI prior to radiation, which is a standard practice in patients with noniodinated SpaceOAR[TM]. It also reduces healthcare costs by circumventing the reimbursement challenges associated with preprocedural MRIs prior to radiation. CT is ubiquitous, inexpensive, and easily available to obtain for postprocedural confirmation. Furthermore, due to the easy visualization of iodine, an MRI for planning may also not be needed [52]. This implies the need for more concrete data on SpaceOAR Vue, along with added benefits, such as lacking the need for a MRI. A long-term study that follows patients over the years could give more insight into the true costs of radiation toxicities with hydrogel placement to better guide decision-making.

## 10. Other Available Spacers

Recently, the US Food and Drug Administration (FDA) has granted clearance for Barrigel[TM], which is made from non-animal-stabilized hyaluronic acid based on a study from a 510(k) premarket submission made to the FDA [53]. A study containing 201 patients including Barrigel[TM] perirectal spacers and a control showed significantly decreased rectal toxicity with IMRT. The primary endpoint of the study was met, which was a 25% reduction in the volume of the rectum, receiving 90% of the prescription radiation dose (lower confidence limit is 0.923, $p < 0.0001$). Barrigel has also been approved for use in Europe and Australia. Although Barrigel shows promise, randomized controlled trials are warranted, and hydrogel spacers have the largest supporting clinical data and are the most widely used to date.

Another prostate rectal spacer that has been extensively studied in preclinical and clinical trials is the ProSpace biodegradable fillable balloon [54,55]. The ProSpace system—a deflated balloon made of biodegradable polymer—is placed perineally with hydrodissection and the use of an introducer. By implanting a balloon, the distance between the prostate and rectum increased by approximately 2 cm that lasts for 6 months. One disadvantage is that the insertion procedure is more invasive than hydrogel or a hyaluronic acid spacer and is performed under general anesthesia [56].

A perineal injection of human collagen prior to IMRT has shown decreased radiation to the rectum [57]. However, the authors described difficulty in obtaining a desired consistency, as the material tends to get lumpy when injected. In addition, the availability of human collagen is volatile, and supply/demand may be dependent on pressures from black markets. In addition, blood patches from 20 mL of the patient's blood were applied to create a prostate–rectal space of 3.9 mm. Although decreased radiation of the rectum was observed during brachytherapy, the distance created is small when compared to other alternatives [58].

## 11. Summary

The hydrogel spacer SpaceOAR has clinically been demonstrated to decrease the radiation dose to the rectum during image-guided IMRT, SBRT, and low-dose rate brachytherapy. In some cases, the long-term follow-up exhibits significant improvements in bowel, urinary, and sexual quality of life, along with significant reductions in late gastrointestinal and genitourinary toxicities. Hydrogels reduce radiation to the rectum by increasing the distance between the prostate and the rectum. This aids in radiation targeting and delivery during EBRT and decreases the radiation dose by the square of the distance during brachytherapy.

SpaceOAR Vue containing iodine is a newer version and has unique advantages when compared to SpaceOAR while preserving its precursor's properties, quality, and robustness. The increase in contrast provides: delineation on planning CT without the need for a MRI, intrafraction treatment accuracy, and an alternative for patients who have contraindications to MRI. In addition, the placement of SpaceOAR Vue is discussed in detail. Regarding adverse effects, both SpaceOAR and SpaceOAR Vue are relatively new to the market, with a good safety profile when inserted appropriately. However, complications have been documented in the literature, usually due to improper placement. Nevertheless, further studies are necessary to understand any serious potential complications.

**Author Contributions:** Writing—original draft preparation, S.R.H. and A.K.G. and writing—review and editing, S.R.H. and A.K.G. All authors have read and agreed to the published version of the manuscript.

**Funding:** This research received no external funding.

**Institutional Review Board Statement:** Not applicable.

**Informed Consent Statement:** Not applicable.

**Data Availability Statement:** Study did not report any data.

**Conflicts of Interest:** The authors declare no conflict of interest.

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
