# Peer review of "Application of Hydrogel Spacer SpaceOAR Vue for Prostate Radiotherapy"

_tomography, doi:10.3390/tomography8060221_

Round 1

Reviewer 1 Report

The manuscript 'Review and application of hydrogel spacers for prostate radiotherapy' by Hadigal and Gupta is focussing on prostate cancer and treatment. It reviews basics on prostate and prostate cancer, treatment regimens, hydrogel spacers, supporting clinical data regarding hydrogels, best practices for Hydrogel placement as well as the risk of adverse effects.

The manuscript is well written and includes all relevant issues. However, a small paragraph on alternative treatment regimens - although they are not focus of the manuscript - would be helpful to allow the reader to have a better overview on current treatments including surgery.

I have no major comments.

Reviewer 2 Report

This paper is focused on the introduction of a new technology, in this case iodinated hydrogel rectal spacer which facilitates radiotherapy planning and may reduce RT toxicity. There are several inacuracies regarding radiotherapy in the manuscript and, moreover, clinical cost-benefit of an invasive procedure added to non-invasive treatment modality is not analysed appropriately. Anyhow, I believe the manuscript may be accepted as soon as following corrections are carried out:

- line 54: IGRT is not considered a distinct type of radiotherapy and is not similar to IMRT. IGRT means frequent use of imaging before or during irradiation, aiming at high accuracy of treatment, while 3D conformal RT, IMRT and VMAT are technologies for radiation dose shaping,

- line 64: increasing the distance between prostate and rectum...decreases radiation toxicity: this notion refering to the inverse square law is relevant for brachytherapy only where the source of irradiation is inside the prostate. For external beam RT simply distancing of an organ at risk (rectum) from PTV is relevant,

-line 70: Hydrogels  - hydrogels,

-line 81: overall target volume -  clinical target volume (volume of the whole prostate identified by CT versus MR),

- line 83: ... significantly reduces rectal toxicity.: This claim must be supported by a publication(s) and specified comparing acute and late graded rectal toxicity without and with a rectal spacer, 

- line 87: ... treatments challenging and lead to inaccuracies in targeted radiation dose. It is uncleare which inaccuracies: incorrect rectum or/and prostate delineation or inacurate positioning/reproducibility?

-line 99: ...despite a smaller evarage delineated volume. Please, explain, smaller in comparison to ...?

- line 114 Quality of Life (QOL) - quality of life (QOL or QoL)

- line 117: ...decreased the incidence of urinary incontinence. Values and significance must be stated.

- line 123: ...meaningful declines in bowel quality... Values and significance must be stated.

- line 129: SBRT radiotherapy - SBRT (Stereotactic Body Radiotherapy)...has greater efficacy: this notion may be misleading as there is no evidence so far for improved tumor control or reduced toxicity by SBRT in comparison to modraterly or conventionally hypofractionation RT. The principle advantage of SBRT is that the curative dose is delivered in just 5 fraction over short period of time,

- lines 131 - 132: periprostatic toxicity... is very vague and undefined term, genitourinary and gastrointestinal toxicity should be used instead,

- line 135: 50 Grays - 50 Gy, 5 patients - out of n=?,

- line 140 ...exposure was decreased by... this is not clear and must be specified: maximum, mean dose or D2 dose or any other?...across the measured dosimetric curve  ... this is not appropriate terminology:  on dose-volume histograms or dosimetric analysis,

- line 143: ...rectal ulcer events... assessed by endoscopy?, provide values and significance,

- line 163: ...distance - proximity,

- line 183: NPO means.... (check also other abbreviations),

- line 194 This portion... e.g. We desribe here... Following....

- line 209: anti-dependent position - ?,

- line 266: Radiation Oncology community - radiation oncology community,

- line 285: ...association...at best unclear .... this is not SAE causality term: must be either unrelated or unlikely causal relationship, 

- line 290: ...controlled study [43] ...this study must be cited and must be traceable in a publications database (!). Please, provide or change comments,

- Conclusion or Summary is missing, pelase complete,

- the role of prostate fiducials markers with rectal spacer has not been mentioned,

- consider SpaceOAR and SpaceOAR CT (transversal and sagital) CT image comparison.

Reviewer 3 Report

The use of rectal spacers in radiotherapy is of great importance. But the authors focus on the spacers by one manufacturer. That makes the article of less use to the community. The authors should be stimulated to present an overview of the most widely used spacers.

Round 2

Reviewer 2 Report

The paper should be finally checked by a radiation oncologists to correct the common radiotherapy terminology once again as there are persistent inaccuracies. 

At least following corrections must be peformed:

- lines 65-72: this is a trivial information with no clear context - should be omitted.

- lines 170 - 183: the text is a real mess: read carefully an re- write in a logical order; which subgroups? significance?

- line 178 ..significant reduction ...in smaller prostates? The abstract [26] states similar reduction in both small and large prostates. Correct the evidence.

- line 189 .... received 45-50 Gy... add: ... dose escalated 45 - 50 Gy as that was  experimental dose escalation study and standard doses are much lower so the relevance for the standard clinical practice is unclear.

-line 373: ...and cost effective in a physician’s office... please explain 

- line 378: ... in high SBRT but not in low SBRT... this is not a correct term, if a  dose level is intended the dose for "low dose" and "high does" must be specified as there is no universal consensus (or use standard dose and dose-escalated SBRT)

- line 410 ... invasive that hydrogel ... than...

.

-

Reviewer 3 Report

The change of the title creates a good match between the title and the content. But my major problem is the  content, not the title. It would be of great value if more available spacers were compared
